# Two New Species of Filamentous Sulfur Bacteria of the Genus *Thiothrix*, *Thiothrix winogradskyi* sp. nov. and ‘*Candidatus* Thiothrix sulfatifontis’ sp. nov.

**DOI:** 10.3390/microorganisms10071300

**Published:** 2022-06-27

**Authors:** Nikolai V. Ravin, Simona Rossetti, Alexey V. Beletsky, Vitaly V. Kadnikov, Tatyana S. Rudenko, Dmitry D. Smolyakov, Marina I. Moskvitina, Maria V. Gureeva, Andrey V. Mardanov, Margarita Yu. Grabovich

**Affiliations:** 1Institute of Bioengineering, Research Center of Biotechnology of the Russian Academy of Sciences, 119071 Moscow, Russia; nravin@biengi.ac.ru (N.V.R.); mortu@yandex.ru (A.V.B.); vkadnikov@bk.ru (V.V.K.); mardanov@biengi.ac.ru (A.V.M.); 2Water Research Institute, IRSA-CNR, Monterotondo, 00185 Rome, Italy; simona.rossetti@irsa.cnr.it; 3Department of Biochemistry and Cell Physiology, Voronezh State University, Universitetskaya pl., 1, 394018 Voronezh, Russia; ipigun6292@gmail.com (T.S.R.); songolifreya@mail.ru (D.D.S.); moskvitina.m04.20@gmail.com (M.I.M.); maryorl@mail.ru (M.V.G.)

**Keywords:** *Thiothrix*, genome, MAG, ANI, dDDH

## Abstract

The metagenome of foulings from sulfidic spring “Serovodorodny” (Tatarstan, Russia), where members of the genus *Thiothrix* was observed, was sequenced. Representatives of the phyla *Gammaproteobacteria*, *Cyanobacteria* and *Campilobacteriota* dominated in the microbial community. The complete genome of *Thiothrix* sp. KT was assembled from the metagenome. It displayed 93.93–99.72% 16S rRNA gene sequence identity to other *Thiothrix* species. The average nucleotide identity (ANI) и digital DNA-DNA hybridization (dDDH) showed that the genome designated KT represents a new species within the genus *Thiothrix*, ‘*Candidatus* Thiothrix sulfatifontis’ sp. nov. KT. The taxonomic status has been determined of the strain *Thiothrix* sp. CT3, isolated about 30 years ago and not assigned to any of *Thiothrix* species due to high 16S rRNA gene sequence identity with related species (i.e., 98.8–99.4%). The complete genome sequence of strain CT3 was determined. The ANI between CT3 and other *Thiothrix* species was below 82%, and the dDDH values were less than 40%, indicating that strain CT3 belongs to a novel species, *Thiothrix winogradskyi* sp. nov. A genome analysis showed that both strains are chemo-organoheterotrophs, chemolithotrophs (in the presence of hydrogen sulfide and thiosulfate) and chemoautotrophs. For the first time, representatives of *Thiothrix* showed anaerobic growth in the presence of thiosulfate.

## 1. Introduction

One of the most interesting groups of filamentous colorless sulfur bacteria are members of the genus *Thiothrix*, first described by Winogradsky in 1888. Representatives of this genus have similar morphotypes, lead an attached lifestyle, form rosettes, mucous sheaths (only *Thiothrix*
*unzii* A1^T^ lacks a mucous sheath) and are able to accumulate elemental sulfur intracellularly during lithotrophic growth in the presence of reduced sulfur compounds [1].

*Thiothrix* species are widespread in sulfidic springs and wastewater treatment plants [2], where their overgrowth may cause activated sludge bulking [3,4]. On the other hand, they may contribute to the removal phosphorus from wastewater [5,6].

Representatives of the genus *Thiothrix* were found in metagenomes from microbial communities associated with plants and animals: in the gill chambers of the deep-sea shrimp *Rimicaris exoculato* [7], in the rhizosphere of the aquatic plant *Eichhornia crassipes* from the Hindon River in India [8], and in the microbiome of a *Chironomus ramosus* larva in the Mutha River, India [9]. Additionally, in Yellowstone Lake [10], in sediments in the North and Baltic Seas [11], in communities from abandoned gold and uranium mines in Poland [12], in underground sulfide sources in a cave in the USA [13], in the system of shallow hydrothermal vents off Kueishantao Islet, Taiwan [14], representatives of the genus *Thiothrix* were found during metagenomic sequencing. Metagenome analyses led to the discovery of members of the genus *Thiothrix* in microbial communities in wastewater treatment plants [15], in sewers [16] and in fermenters simulating wastewater treatment systems [17,18,19,20,21,22].

Species retained in the genus *Thiothrix* after the last taxonomic revision [23] included *Thiothrix nivea*, *Thiothrix fructosivorans*, *Thiothrix unzii*, *Thiothrix caldifontis*, and *Thiothrix lacustris*. The strain *Thiothrix* sp. CT3 also belongs to this genus. Genome sequences for *T. nivea* JP2^T^ (GCF_000260135.1), *T. lacustris* BL^T^ (GCF_000621325.1), and *T. caldifontis* G1^T^ (GCF_900107695.1) were obtained in 2012, 2014, and 2016. Later, in 2020–2021, whole genome sequences of type strains were obtained for *T. unzii* A1^T^ (ATCC 49747^T^) and *T. fructosivorans* Q^T^ (ATCC 49748^T^), as well as for new isolates and MAG’s [24,25,26]. Thus, over the last few years, the genus *Thiothrix* has expanded to 10 species. Phylogenetic reconstructions based on genomes and pairwise ANI and dDDH values clearly confirmed the independence of the species *T. nivea*, *T. caldifontis*, *T. lacustris*, *T. fructosivorans* and *T. unzii*, and showed that *Thiothrix* sp. AS, *Thiothrix* sp. Ku-5, MAGs of *Thiothrix* sp. A52, *Thiothrix* sp. RT and *Thiothrix* sp. SSD2 represent new species, for which the following names were proposed: *Thiothrix litoralis* AS^T^ sp. nov., *Thiothrix subterranea* Ku-5^T^ sp. nov., ‘*Candidatus* Thiothrix anitrata’ sp. nov. A52, ‘*Candidatus* Thiothrix moscowensis’ sp. nov RT, and ‘*Candidatus* Thiothrix singaporensis’ sp. nov. SSD2 [25,26].

It was shown for many isolates from the genus *Thiothrix*, that the standard phylogenetic marker, the 16S rRNA gene, is not informative due to very high sequence identity [23,26,27]. The 16S rRNA gene sequence identity between *Thiothrix* sp. CT3 and phenotypically and phylogenetically close species *T. fructosivorans* Q^T^, *T. caldifontis* G1^T^, and *T. lacustris* BL^T^ was shown to be 99.4, 98.8, and 98.9%, respectively. Such high identity did not allow the identification of *Thiothrix* sp. CT3 at the species level [28,29]. Likewise, the taxonomic identification of another strain, *Thiothrix* sp. AS, on the basis of 16S rRNA gene sequence, suggested that it belongs to the species *T. lacustris* (100% identity). However, the phylogenetic analysis based on a whole genome sequence showed that this strain represents a distinct species, *T. litoralis* sp. nov. [26]. Due to the current lack of reliable phylogenetic markers only the analysis of the whole genome sequences can clarify the phylogenetic status of representatives of the genus *Thiothrix*.

*Thiothrix* sp. CT3 was isolated from an activated sludge treatment plant in 1994 in Bari West, Italy (41°06′41″ N 16°51′19″ E) [30]. The phenotypic characteristics of *Thiothrix* sp. CT3 were studied in detail by Rossetti et al. (2003). Later, the major fatty acids were determined for this strain [27]; however, until now, its taxonomic status has remained unclear due to the lack of a whole genome sequence, the similarity of its phenotypic characteristics and its high 16S rRNA gene sequence identity with phylogenetically close *Thiothrix* species. 

We determined the complete genome of *Thiothrix* sp. CT3 and complete sequence of MAG of *Thiothrix* sp. KT from bacterial fouling formed at the outflow of water from the “Serovodorodny” (Серoвoдoрoдный) spring in Tatarstan, Russia (54°30′22.9″ N 52°09′29.4″ E). Based on genome sequences, we have described two new species of the genus *Thiothrix*: *Thiothrix winogradskyi* sp. nov. CT3 and ‘*Candidatus* Thiothrix sulfatifontis’ sp. nov. KT.

## 2. Materials and Methods

### 2.1. Cultivation of Thiothrix sp. CT3

Strain *Thiothrix* sp. CT3 (=DSM 12730^T^ = VKM B-3582^T^) was analyzed in this study. A nutrient medium of the following composition was used for its cultivation (gL^−1^): (NH_4_)_2_SO_4_ (0.5), CaCl_2_ (0.03), KH_2_PO_4_ (0.11), K_2_HPO_4_ (0.085), MgSO_4_·7H_2_O (0.1), NaNO_3_ (0.3), BD Difco™ Agar, Technical (0.5), NaHCO_3_ (0.5 gL^−1^ (6 mM)), Na_2_S_2_O_3_·5H_2_O (1 gL^−1^ (4 mM)), the trace of vitamins and microelements [31]. NaHCO_3_ was autoclaved separately as a 10% solution and added to the culture medium after autoclaving.

Physiological tests were carried out using a medium of the above composition with alternative carbon sources. Three passages of culture were carried out with each of the substrates to test bacterial growth [27].

### 2.2. Sampling Site Used for Obtaining MAG of Thiothrix sp. KT and Its Physicochemical Characteristics

The biomass sample for metagenomic analysis was taken from microbial fouling in the water flow from the “Serovodorodny” spring in Tatarstan, Russia.

The physicochemical parameters of the water (pH, temperature and redox potential) were measured with an HI18314F pH meter (Hanna Instruments, Vöhringen, Germany). The concentration of acid-labile sulfide in the samples was determined by the spectrophotometric method with paraphenylenediamine, and by the method of direct iodometric titration, preliminarily fixing the sulfide with 10% zinc acetate. The concentration of dissolved oxygen in the medium was determined using a Hi 9142 oxygen meter (Romania). The total mineralization was determined by the method of electrical conductivity using a Multitest KSL-101 conductometer.

### 2.3. Thiothrix sp. CT3 Genome Sequencing

Genomic DNA was isolated from *Thiothrix* sp. CT3 using a DNeasy PowerSoil DNA isolation kit (Mo Bio Laboratories, Carlsbad, CA, USA) and sequenced using Illumina and Oxford Nanopore technologies. For Illumina sequencing, the shotgun genome library was prepared using the NEBNext Ultra II DNA library prep kit (New England Biolabs, Ipswich, MA, USA) and sequenced on an Illumina MiSeq instrument in a paired reads mode (2 × 300 nt). A total of 872,344 reads pairs were generated. Low quality sequences were trimmed using Sickle v.1.33 (q = 30). Since *Thiothrix* species usually contains multiple copies of rRNA operons, in order to obtain complete closed genome, genomic DNA was additionally sequenced on a MinION device (Oxford Nanopore Technologies, Oxford, UK) using the ligation sequencing kit 1D and FLOMIN110 cells. 69,826 reads with an average length of 5054 nt were obtained. MinION reads were assembled into three circular contigs using Flye v. 2.8.2 [32]. The consensus sequence of the assembled contigs was corrected with two iterations of Pilon v.1.22 [33] using Illumina reads.

### 2.4. 16S rRNA Gene Fragment Sequencing and Analysis

The total DNA was extracted from 200 mg of a microbial fouling of the “Serovodorodny” spring (Karkaly, Tatarstan, Russia) using a DNeasy PowerSoil DNA isolation kit (Qiagen, Hilden, Germany). 16S rRNA gene fragments comprising the V3–V4 variable regions were obtained by PCR amplification with the universal primers 341F (5′-CCTAYG GGDBGCWSCAG) and 806R (5′-GGA CTA CNVGGG THTCTAAT) [34]. The obtained PCR fragments were bar-coded using the Nextera XT Index Kit v. 2 (Illumina, San Diego, CA, USA) and sequenced on Illumina MiSeq (2 × 300 nt reads). Pairwise overlapping reads were merged using FLASH v.1.2.11 [35]. Obtained sequences were clustered into operational taxonomic units (OTUs) at 97% identity threshold using the USEARCH v. 11 program [36]. Low quality reads were removed prior to clustering, and chimeric sequences were removed during clustering by the USEARCH algorithms. To calculate the relative abundancies of OTUs, all obtained reads (including low-quality reads) were mapped to OTU sequences at a 97% global identity threshold by USEARCH. OTUs composed of only a single read were discarded. The taxonomic assignment of OTUs was performed by searching against the SILVA v.138 rRNA sequence database using the VSEARCH v. 2.14.1 algorithm [37]. OTUs assigned to chloroplasts were excluded from the analysis.

### 2.5. Metagenome Sequencing and Assembly of Thiothrix sp. KT MAG

The total DNA was extracted from microbial fouling of the “Serovodorodny” spring (Tatarstan, Russia) as described in point 2.4. Metagenome sequencing of total DNA isolated from microbial fouling of the “Serovodorodny” spring was performed using Illumina and Oxford Nanopore platforms. The library for Illumina sequencing was prepared using the NEBNext Ultra II DNA library preparation kit (New England Biolabs, Ipswich, MA, USA) and sequenced on Illumina MiSeq instrument in a paired-end (2 × 300 bp) mode. The sequencing generated 3,154,438 read pairs (~1.9 Gb). Trimming of low-quality sequences (Q < 30) was performed using Sickle v.1.33. Metagenomic DNA was also sequenced using Oxford Nanopore platform (MinION instrument, Oxford Nanopore Technologies, Oxford, UK) as described above; 158,993 reads with an average length of 3633 nt were obtained. MinION reads were assembled into contigs using Flye v. 2.9 [13].

The obtained contigs were binned into MAGs using CONCOCT v.1.1.0 [38]. The obtained MAGs were taxonomically classified using the Genome Taxonomy Database Toolkit (GTDB-Tk) v.1.5.0 [39] and Genome Taxonomy database (GTDB) [40]. One of the obtained MAGs, assigned to the genus *Thiothrix*, was manually assembled into a single circular contig using the Flye assembly graph visualized in Bandage v. 0.8.1 tool [41]. The assembly sequence was corrected using Illumina reads with two iterations of Pilon v. 1.22 [17].

### 2.6. Annotation and Analysis of the Genomes

The search for genes and their annotation was carried out using RAST server 2 [42], followed by manual correction of the annotation by comparing the predicted protein sequences with the National Center for Biotechnology Information (NCBI) databases. The *N*-terminal signal peptides were predicted by Signal P v.5.0 (https://services.healthtech.dtu.dk/service.php?SignalP-5.0 (accessed on 10 February 2022)), and the presence of transmembrane helices was predicted by TMHMM v.2.0 (http://www.cbs.dtu.dk/services/TMHMM/ (accessed on 10 February 2022)).

ANI was calculated using an online resource (https://www.ezbiocloud.net/tools/ani (accessed on 10 February 2022)) based on the OrthoANIu algorithm [31]. dDDH calculation was performed using the GGDC online platform (https://ggdc.dsmz.de/ggdc.php# (accessed on 8 November 2021)). Pairwise 16S rRNA gene sequence alignments were performed using the Smith-Waterman alignment algorithm (https://www.ebi.ac.uk/Tools/psa/emboss_water/ (accessed on 10 February 2022)).

### 2.7. Phylogenetic Analysis

The dataset used for genome-based phylogenetic analysis included two newly sequenced *Thiothrix* genomes along with genomes of *T. litoralis* AS^T^, *T. lacustris* BL^T^, *T. fructosivorans* Q^T^, *T. caldifontis* G1^T^, *T. subterranea* Ku-5^T^, ‘*Ca*. Thiothrix anitrata’ A52, *T. unzii* A1^T^, ‘*Ca*. Thiothrix moscowensis’ RT, ‘*Ca*. Thiothrix singaporensis’ SSD2, *T. nivea* JP2^T^, *Thiolinea disciformis* B3-1^T^, *Thiothrix (Thiolinea) eikelboomii* AP3^T^, *Thiofilum flexile* EJ2M-B^T^, and *Leucothrix mucor* DSM 2157^T^. The GTDB-Tk was used to identify the 120 single-copy, phylogenetically informative conservative marker genes in these genomes. The multiple alignment of concatenated amino acid sequences of these genes, obtained in GTDB-Tk, was used to construct a maximum-likelihood phylogenetic tree in PhyML v. 3.3 [43] with default parameters (LG+G evolutionary model). The level of support for internal branches was assessed using a Bayesian test in PhyML.

### 2.8. Experimental Verification of Thiosulfate Reductase Functioning

The amount of protein was determined by the Lowry method [44]. Separate determination of S_2_O_3_^2−^, SO_3_^2−^ and HS^−^ in the medium was carried out by the method of separate iodometric titration [45].

## 3. Results

### 3.1. Biotope Harboring Thiothrix sp. KT: Physicochemical Characteristics and Microbial Community Composition

The sample for metagenomic analysis was taken from microbial fouling at the outflow of water in the sulfidic spring “Serovodorodny” in Tatarstan, Russia (Appendix A). The water temperature at the sampling site was 15–18 °C, pH 7.0. The total salinity of the water was 3.5–5.0 g/L, while the concentration of sulfates was 2.9 g/L. The concentrations of hydrogen sulfide and oxygen were in the range 2–4 mg/L and 1.4–5.0 mg/L respectively.

To characterize the composition of microbial fouling, 7685 sequences of 16S rRNA gene fragments were determined. As a result of clustering the obtained sequences, 164 OTUs, all of which were assigned to bacteria, were identified. The microbial community was dominated by the *Proteobacteria* (46.1% of all 16S rRNA gene sequences) of classes *Alpha-*(5.4%) and *Gammaproteobacteria* (40.7%), *Cyanobacteria* (22.7%), *Campylobacterota* (20.4%) and *Bacteroidota* (6.8%) (Figure 1). The most abundant physiological group was sulfur-oxidizing bacteria—*Gammaproteobacteria* of the genus *Thiothrix* (27.1%), *Campylobacterota* of the genus *Sulfovorum* (20.2%), and unclassified *Gammaproteobacteria* of the family *Halothiobacillaceae* (9.5%). *Cyanobacteria* were dominated by a single OTU phylogenetically close to the genus *Microcoleus* (98.65% identity to *M. autumnalis* BEA 1075B). *Microcoleus* sp. are widespread in freshwater environments where they form cohesive mats, which can cover large areas in lakes and rivers [46]. The rest of the community consisted of heterotrophic *Alphaproteobacteria* and *Bacteroidota*, which can utilize organic substances formed by primary producers—cyanobacteria and chemolithoautotrophic sulfur oxidizers.

### 3.2. Phenotypic Characteristics of Thiothrix sp. CT3

The data of physiological characteristics presented in this study were obtained by us earlier [26,27]. Data for *Thiothrix* sp. CT3, obtained by [29], are confirmed in this study. A comparison of the phenotypic properties of *Thiothrix* sp. CT3 with other members of *Thiothrix* from clade 1 is shown in Table 1.

### 3.3. General Genome Properties

The genome of *Thiothrix* sp. CT3 was assembled into three contigs: one circular 4,343,903 bp long chromosome and two circular plasmids with lengths of 20,671 bp and 15,872 bp. Genome annotation of *Thiothrix* sp. CT3 identified five copies of the 16S-23S-5S rRNA operon, 66 tRNA genes, and 4292 potential protein-coding genes. The G+C mol% of the genome is 51.4%. MAG of *Thiothrix* sp. KT was obtained as a circular chromosome with a length of 3,692,320 bp. The genome annotation revealed two copies of the 16S-23S-5S rRNA operon, 41 tRNA genes and 3729 potential protein-coding genes. The G+C content of the genome is 51.5 mol%. The general characteristics of the newly sequenced genomes of the genus *Thiothrix* are shown in Table 2.

### 3.4. Phylogenetic Analysis

The levels of 16S rRNA gene sequence identity between *Thiothrix* sp. CT3 and other members of the genus *Thiothrix* varied from 94.98 to 99.37%. ANI values between *Thiothrix* sp. CT3 and other *Thiothrix* genomes were less than 81.43%, which is below the species delineation threshold of 95% (Figure 2) [30]. The ANI data were also consistent with the results of dDDH (<40.10%) (Appendix A). The obtained genome-to-genome distance values allowed the assignment of *Thiothrix* sp. CT3 to a new species. We propose a name for this new species, *Thiothrix winogradskyi* sp. nov. CT3^T^, after Prof. Sergey Winogradsky (1856–1953), who described the genus *Thiothrix*.

The sequence identities of the 16S rRNA genes of *Thiothrix* sp. KT with other members of the genus *Thiothrix* were in the range 93.93–99.72% (Figure 2). However, the ANI values between MAG of *Thiothrix* sp. KT and other *Thiothrix* genomes were below 91%, and the dDDH values ranged from 22.90 to 48.70% (Appendix A). These values allowed us to classify MAG of *Thiothrix* sp. KT as a novel species within the genus *Thiothrix*, for which we propose the specific epithet ‘*Candidatus* Thiothrix sulfatifontis’ sp. nov. KT. The organism received its name due to its development in a biotope where the concentration of sulfates was 2.9 g/L, while other members of the genus *Thiothrix* thrived in biotopes where the concentration of sulfates was an order of magnitude lower.

The phylogenetic position of new *Thiothrix* species, ‘*Ca.* Thiothrix sulfatifontis’ KT and *T. winogradskyi* CT3^T^, along with the previously described species, was also analyzed by constructing a phylogenetic tree based on concatenated sequences of 120 conserved marker genes (Figure 3). New genomes appeared to form distinct species-level lineages within the genus *Thiothrix*, and were most closely related to *T. subterranea* Ku-5^T^, isolated from the outflow of sulfidic water from a drained well from the closed and flooded coal mine in Siberia [26]. These three species, along with *T. litoralis* AS^T^, *T. lacustris* BL^T^, *T. fructosivorans* Q^T^, and *T. caldifontis* G1^T^ form a distinct Clade 1. The same cluster appeared on the 16S rRNA gene-based maximum-likelihood phylogenetic tree, but phylogenetic relationships within this clade were poorly resolved, as indicated by the lower level of support for internal branches (Appendix A).

### 3.5. Description of Thiothrix winogradskyi sp. nov.

*T. winogradskyi* (wi.no.grad′sky.i. N.L. masc. gen. n. *winogradskyi* of Winogradsky, named after Prof. Sergey Winogradsky (1856–1953), who described the genus *Thiothrix*). 

Rod-shaped cells with rounded ends, united in multicellular filaments (trichomes) with polysaccharide sheaths. Gram-stain-negative and aerobic. Cells of the major form are 0.8–2.0 μm in diameter and 4.3–6.7 μm long. Filaments are non-motile. Gliding gonidia are produced from the apical ends of the filaments. Gonidia can form rosettes. The temperature range for growth is 10–30 °C, with optimum growth at 20–24 °C. The pH range for growth is 7.0–8.0, with optimum growth at pH 7.6. Capable of litho-autotrophic growth in the presence of reduced sulfur compounds, as well as heterotrophic growth. Uses acetate, pyruvate, succinate and lactate as carbon sources, and does not use malate, oxaloacetate, sucrose, fructose, propionate, mannose, trehalose, TWEEN^®^ 80, phenol, butyrate, n-amyl alcohol, oleic acid, benzoate, butanol, D-(+)-glucose, lactose, maltose, citrate, ethanol, methanol, propanol and glycerol. The CT3 strain is capable of using ammonium, nitrite, nitrate and urea as nitrogen sources. Oxidase-positive, catalase-negative. Capable of dissimilatory nitrate reduction: nitrate is reduced to N2O. It stores polyhydroxybutyrates and sulfur intracellularly. Oxidizes thiosulfate to sulfate and elemental sulfur [29]. The main fatty acids are C_16:1ω7_ (45.7%), C_16:0_ (18%) и C_18:1ω7_ (34.2%) [27].

The type strain is CT3^T^ (=DSM 12730^T^ = VKM B-3582^T^) isolated from treatment plants in Bari, Italy. The G+C content of the DNA of the type strain is 51.4%. The GenBank accession number for the 16S rRNA gene sequence and the genome sequence of the type strain are AF148516.1 and GCA_021650935.1, respectively.

### 3.6. Description of ‘Ca. Thiothrix sulfatifontis’ sp. nov.

*‘Ca.* Thiothrix sulfatifontis’ (sul.fa.ti′fon.tis. N.L. masc. n. *sulfas*, *sulfatis* sulfate; L. n. *fons*, *fontis* a spring; N.L. gen. n. *sulfatifontis* from a sulfate spring, pertaining to the source of obtaining a metagenome). Not cultivated. Inferred to be a rod-shaped, non-motile, aerobic litho-autotroph. Obtains energy by oxidation of reduced sulfur compounds and fix carbon in the Calvin cycle. Not capable of dissimilatory nitrate reduction and fixation of dinitrogen gas. Represented by MAG obtained from the metagenome of bacterial fouling formed in the water flowing from the sulfidic sulfate-rich spring in Tatarstan, Russia.

MAG was obtained from hydrogen sulfide source “Serovodorodny”, Tatarstan, Russia. The G+C content of the DNA is 51.5%. The GenBank accession number for the genome sequence is GCA_022828425.1, respectively.

### 3.7. Genome Analysis

A genome analysis of *T. winogradskyi* CT3^T^ and ‘*Ca*. Thiothrix sulfatifontis’ KT revealed a complete set of genes for the electron transfer chain and the Krebs cycle, indicating a respiratory type of metabolism (Appendix A). The formation of pyruvate from D-(+)-glucose in both organisms is possible via the Embden-Meyerhof-Parnas pathway and the pentose phosphate pathway.

Genomes of *T. winogradskyi* CT3^T^ and ‘*Ca*. Thiothrix sulfatifontis’ KT encode the complete set of genes of the Calvin cycle. Both genomes contained genes encoding Forms IAc and II RuBisCO (ribulose 1,5-bisphosphate carboxylase/oxygenase), while ‘*Ca*. Thiothrix sulfatifontis’ KT additionally encodes Form IAq enzyme. The presence of genes of the Calvin cycle in the *T. winogradskyi* CT3^T^ genome is consistent with the ability of this strain to grow autotrophically [29].

Both genomes contain genes *nasA* for assimilatory nitrate reductase. No genes for dissimilatory reduction of nitrogen compounds were found in the genome of ‘*Ca*. Thiothrix sulfatifontis’ KT. In contrast, the genome of *T. winogradskyi* CT3^T^ contains genes *narGHI* for the membrane-bound dissimilatory nitrate reductase. The reduction of nitrite to gaseous nitric oxide (NO) could be carried out by nitrite reductase (NirS). The gene for NO reductase (Nor), *cnorBC*, which catalyzes the reduction of NO to nitrous oxide (N_2_O), were also present in the genome of *T. winogradskyi* CT3^T^. The *nosZ* gene for N_2_O reductase (Nos), which reduces N_2_O to molecular nitrogen, was absent. Thus, *T. winogradskyi* CT3^T^ is likely capable of respiration with nitrate, nitrite and NO. Genome-based predictions are consistent with the observed ability of *T. winogradskyi* CT3^T^ to reduce nitrate to nitrite [29]. Neither strains can fix gaseous nitrogen, as indicated by the lack of genes for catalytic subunits of nitrogenase (*nifHDK*).

Dissimilatory sulfur metabolism in *T. winogradskyi* CT3^T^ and ‘*Ca*. Thiothrix sulfatifontis’ KT is presented by several enzymatic systems. The genomes of both species contain the genes *fccAB* for flavocytochrome *c*-sulfide dehydrogenase and *sqrAF* for sulfide:quinone oxidoreductase, which are necessary for the oxidation of sulfide to sulfur/polysulfide. Both genomes encoded the *soxAXBYZ* genes of the branched Sox pathway for the oxidation of thiosulfate to sulfur and sulfate. For the *T. winogradskyi* CT3^T^ strain, it was experimentally shown that the oxidation products of thiosulfate are sulfur and sulfate [29]. The oxidation of elemental sulfur to sulfite can be carried out under the action of the reverse Dsr (rDSR) complex (*dsrABEFHCMKLJOPNRS*). The oxidation of sulfite to sulfate can be carried out via a directly by sulfite: quinone oxidoreductase (*soeABC*), or via indirect pathway by APS reductase (*aprAB*) and ATP sulfurylase (*sat*). A genome analysis of ‘*Ca*. Thiothrix sulfatifontis’ KT revealed genes encoding membrane-bound molybdopterin oxidoreductase of the Psr/Phs family: *phsA*, which encodes a catalytic subunit with molybdopterin in the active center; *phsB*, which encodes an electron transfer subunit with the [Fe–S] cluster; and *phsC*, which encodes a membrane anchor protein, participating in the transfer of electrons from the quinone pool. The catalytic PhsA subunit contains the N-terminal Tat signal peptide indicating its location on the outer side of the membrane. Such complexes could be responsible for the reduction of various electron acceptors, including thiosulfate, tetrathionate, nitrate and arsenate [46]. Besides ‘*Ca*. Thiothrix sulfatifontis’ KT genes encoding potential thiosulphate reductases were found in the genomes of other members of the genus *Thiothrix*: *T. fructosivorans* Q^T^, *T. caldifontis* G1^T^, *T. lacustris* BL^T^, *T. litoralis* AS^T^, *T. nivea* JP2^T^, *T. unzii* A1^T^, MAG of *Thiothrix* sp. 207, “*Ca*. Thiothrix anitrata” A52, *T. subterranea* Ku-5^T^. For some members of the genus, anaerobic growth on thiosulfate (about 15 mg protein/l) was shown with the formation of sulfide and sulfite, products characteristic of the reduction of thiosulfate with the participation of thiosulfate reductase: *T. caldifontis* G1^T^, *T. lacustris* BL^T^, *T. litoralis* AS^T^, *T. nivea* JP2^T^, *T. unzii* A1^T^.

## 4. Discussion

Unique, unusual habitats are often a source of new, previously undescribed isolates. An analysis of the metagenome of the microbial community of sulfidic spring “Serovodorodny” (Tatarstan, Russia) showed that its composition is typical for microbial mats of a sulfide spring developing under natural light, i.e., the predominance of proteobacteria and cyanobacteria in the above-ground sulfur microbial mats and the dominance of bacteria of the genus *Thiothrix* among proteobacteria [10]. However, it should be noted that this biotope is characterized by a high concentration of sulfates (2.9 g/L), which is not typical for biotopes of representatives of the genus *Thiothrix*. This was probably an unusual factor for the formation of a new species of the genus. It was possible to assemble the MAG *Thiothrix* sp. KT from the metagenome of microbial fouling of the “Serovodorodny” spring, which was assigned to a new species, ‘*Ca*. Thiothrix sulfatifontis’ sp. nov. KT, according to its ANI and dDDH values.

In addition, during this work a whole genome sequence of the strain *Thiothrix* sp. CT3 was obtained. Its analysis made it possible to describe the strain as a representative of a new species—*Thiothrix winogradskyi* sp. nov. CT3.

In both genomes, genes encoding enzymes of the main metabolic pathways of energy and constructive metabolism, providing organo-heterotrophic, chemolithoheterotrophic and chemolithoautotrophic growth, were found. However, metabolic features have been identified. In particular, the ‘*Ca*. Thiothrix sulfatifontis’ KT does not encode genes for dissimilation nitrate reduction, which was previously shown only for ‘*Ca*. Thiothrix anitrata’ [26]. Also, an analysis of the genome sequences of representatives of the genus *Thiothrix*, currently available in GeneBank, made it possible, for the first time, to identify the *phsABC* genes of dissimilatory thiosulfate reductase in *T. fructosivorans* Q^T^, *T. caldifontis* G1^T^, *T. lacustris* BL^T^, *T. litoralis* AS^T^, *T. nivea* JP2^T^, *T. unzii* A1^T^, MAG of *Thiothrix* sp. 207, ’*Ca*. Thiothrix anitrata’ A52, *T. subterranea* Ku-5T, ‘*Ca*. Thiothrix sulfatifontis’ KT, except for *T. winogradskyi* CT3^T^ and ‘*Ca*. Thiothrix moscowensis’ RT. Representatives of the genus Thiothrix, isolated in pure culture and in which the *phsABC* genes encoding thiosulfate reductase were found, were capable of anaerobic growth in the presence of thiosulfate.

## 5. Conclusions

This study describes two new species within the genus *Thiothrix*: ‘*Candidatus* Thiothrix sulfatifontis’ sp. nov. KT and *Thiothrix winogradskyi* sp. nov. CT3^T^. 

MAG *Thiothrix* sp. KT was assembled from the metagenome of the microbial community of the “Serovodorodny” sulfidic spring (Tatarstan, Russia).

For more than 30 years, *Thiothrix* sp. CT3 was not determined as a species due to the low information content of its 16S rRNA gene as a phylogenetic marker. In this work, the complete genome sequence of strain CT3 was determined. Phylogenomic analysis and the use of similarity indexes for genomic comparisons, i.e., AAI, dDDH and ANI, made it possible to assign strain CT3 to a new species, namely, *Thiothrix winogradskyi*.

## Figures and Tables

**Figure 1 microorganisms-10-01300-f001:**
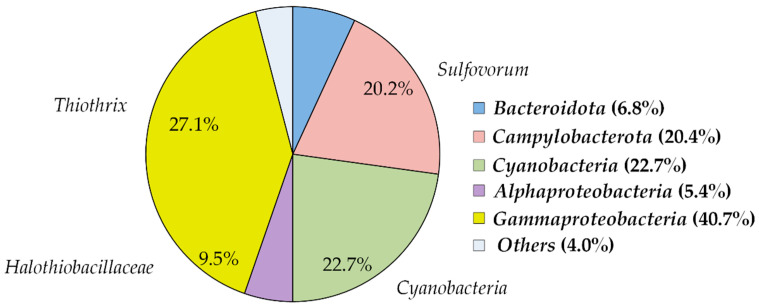
Prokaryotic community composition in “Serovodorodny” spring according to the results of 16S rRNA gene profiling. The community composition is shown at the phylum level, with the exception of *Proteobacteria*, for which the classes *Alpha*- and *Gammaproteobacteria* are shown. The relative abundancies of sulfur-oxidizing groups are shown inside the diagram and highlighted in red.

**Figure 2 microorganisms-10-01300-f002:**
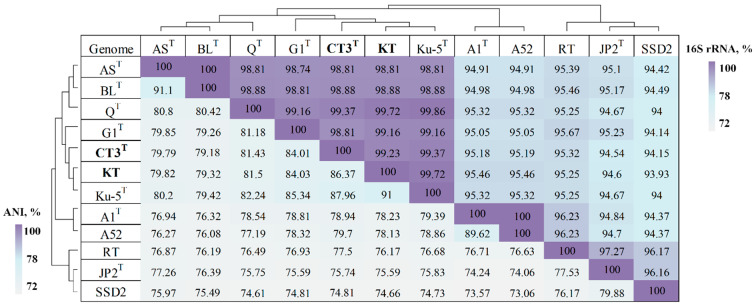
16S rRNA and ANI pairwise values for *Thiothrix* genomes. *T. lacustris* BL^T^, (GCF_000621325.1); *T. litoralis* AS^T^ (GCF_017901135.1); *T. subterranea* Ku-5^T^ (GCF_016772315.1); MAG of *Thiothrix* sp. KT (GCA_022828425.1); *T. caldifontis* G1^T^ (GCF_900107695.1); *Thiothrix* sp. CT3 (GCA_021650935.1); *T. fructosivorans* Q^T^ (GCA_017349355.1); *T. unzii* A1^T^ (GCA_017901175.1); ‘*Ca*. Thiothrix anitrata’ A52 (GCF_017901155.1); ‘*Ca*. Thiothrix moscowensis’ RT (GCA_016292235.1); *T. nivea* JP2^T^ (GCF_000260135.1); ‘*Ca*. Thiothrix singaporensis’ SSD2 (GCA_013693955.1).

**Figure 3 microorganisms-10-01300-f003:**
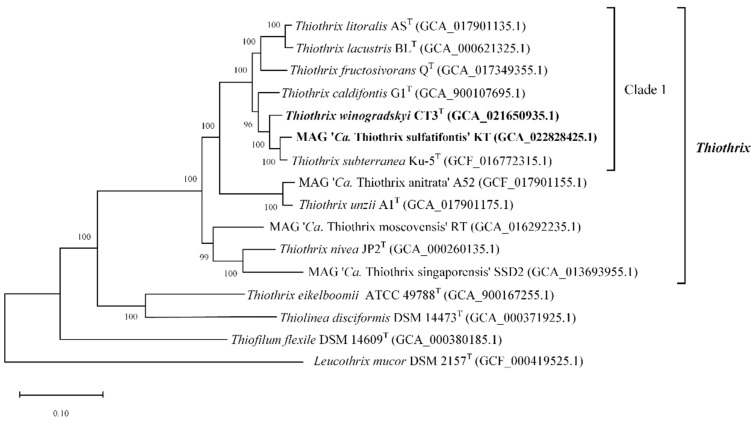
Genome-based phylogenetic tree of type strains of *Thiothrix* species. The GenBank assembly registration number is listed after the genome names. The internal branching support levels assessed by the Bayesian test in PhyML are specified in nodes. The genome of *Leucothrix mucor* DSM 2157^T^ was used for tree rooting. Note that *Thiothrix eikelboomii* actually should belong to the genus *Thiolinea* but cannot be reclassified because of its absence in two international collections [23].

**Table 1 microorganisms-10-01300-t001:** Differentiating characteristics between phylogenetically related species from the genus *Thiothrix* (Clade I). All strains utilize acetate, lactate and pyruvate. Glyoxylate, glycolate, benzoate, salicylate, methanol, isopropanol (propan-2-ol), glycerol, lactose, D-galactose, D-(+)-glucose, D-mannose, serine, lysine, tryptophan, methionine, tyrosine, ornithine, glutamine and alanine do not support the growth of all strains. All strains are capable of chemolitoautotrophic growth with thiosulfate and bisulfide. All strains are capable of respiration of nitrate to nitrite. No strains hydrolyze gelatin or starch. All strains are catalase-negative. None of the strains grow at 3% NaCl. *Thiothrix litoralis* AS^T^ is able to survive with up to 3% NaCl in the medium. The major fatty acids for all strains are C_16:1_^ω7^, C_16:0_, C_18:1_^ω7^. Data are from [26,27].

Characteristic	*Thiothrix* sp. CT3	*T. litoralis* AS^T^	*T. lacustris* BL^T^	*T. fructosivorans* I	*T. caldifontis* G1^T^	*T. subterranea* Ku-5^T^
Natural habitat	activated sludge	seashore of the White Sea	sulfide spring	activated sludge	sulfide spring	sulfide-containing waters from a coal mine
Cell size (µm)	0.8–2.0 × 4.3–6.7	0.8–2.2 × 4.3–6.4	0.9–2.3 × 4.4–6.3	1.0–1.7 × 4.9–10.0	0.9–2.2 × 3.2–6.5	1.19–1.8 × 4.0–6.3
Optimum (range) pH for growth	7.6 (7.0–8.0)	7.4–7.5(6.7–8.0)	7.0 (6.2–8.2)	7.6–8.0 (6.7–8.0)	8.0 (7.0–8.6)	7.4–7.5(6.8–8.0)
Optimum (range) temperature for growth (°C)	20–24 (10–30)	20–22 (4–28)	24 (5–32)	25–27 (5–32)	25 (7–37)	20–22 (4–28)
Organic substrates utilized for growth						
Organic acids:						
Malate	−	−	−	+	−	−
Oxalate	−	−	+	+	−	+
Oxaloacetate	−	−	+	+	+	+
Citrate	−	−	+	−	−	−
Isocitrate	−	−	+	+	−	−
2-oxoglutarate	−	−	+	−	−	−
Formate	−	−	−	+	−	−
Aconitate	−	−	+	−	−	+
Malonate	−	+	+	−	−	+
Succinate	+	+	+	+	+	−
Alcohols:						
Inositol	−	−	−	−	−	+
Ethanol	−	−	−	−	−	+
Butanol	−	−	−	−	−	+
Isobutanol	−	−	−	−	−	+
Mannitol	−	−	−	−	−	+
Sorbitol	−	−	−	−	−	+
Carbohydrates:						
L-Arabinose	−	−	−	+	−	+
D-Xylose	−	−	−	+	−	−
D-Fructose	−	+	−	+	−	+
L-Rhamnose	−	−	−	−	−	+
L-Sorbose	−	−	−	−	−	+
Sucrose	−	−	−	+	−	−
Maltose	−	+	−	+	−	+
Trehalose	−	+	−	−	−	−
Raffinose	−	−	−	+	−	+
Amino acids:						
Isoleucine	−	+	−	−	+	+
Leucine	−	+	−	−	+	+
Proline	−	−	−	−	−	+
Cystein	−	−	+	−	−	−
Asparagine	−	+	+	−	−	−
Phenylalanine	−	−	−	−	−	+
Aspartate	−	+	+	−	+	−
Glutamate	−	+	+	−	−	−
Histidine	−	−	−	−	−	+
Complex media:						
Peptone	−	+	−	−	−	+
Yeast extract	−	+	−	−	−	−
Diazotrophy	−	+	−	−	+	+
Major fatty acids:						
C_16:1_^ω7^, C_16:0_, C_18:1_^ω7^	+	+	+	+	+	+

**Table 2 microorganisms-10-01300-t002:** The general properties of assembled *Thiothrix* genomes.

Species	Genome Assembly	Size (MB)	Contigs	G+C Content (mol %)	Proteins	16S rRNA Gene Copies	tRNAs	Plasmids *
*Thiothrix* sp. CT3 (DSM 12730)	GCA_021650935.1	4.38	3	51.4	4292	5	66	2
MAG of *Thiothrix* sp. KT	GCA_022828425.1	3.69	1	51.5	3729	2	47	NA *

* NA, not applicable for MAGs.

## Data Availability

Genome sequences of *Thiothrix* sp. CT3 and MAG of *Thiothrix* sp. KT has been deposited in NCBI GenBank database under the accession numbers CP091244-CP091246 and CP094685 respectively.

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
