# Peer review of "Two New Species of Filamentous Sulfur Bacteria of the Genus Thiothrix, Thiothrix winogradskyi sp. nov. and ‘Candidatus Thiothrix sulfatifontis’ sp. nov."

_microorganisms, 2022, doi:10.3390/microorganisms10071300_

Round 1
Reviewer 1 Report
The unique and unusual habitats usually represent the main factor for the formation of new species of microorganisms. This is the case of sulfidic spring "Serovodorodny" (Tatarstan, Russia), from where were isolated and analyzed the metagenome of members of the genus Thiothrix.
The aim of this study was to determine the complete genome of Thiothrix sp. CT3 and complete sequence of MAG of Thiothrix sp. KT from bacterial fouling formed at the outflow of water from the "Serovodorodny" (Сероводородный) spring in Tatarstan, Russia (54°30’22.9"N52°09’29.4"E).
Based on pieces of information obtained from genome sequences, the authors described two new species of the genus Thiothrix, Thiothrix winogradskyi sp. nov. CT3 and ‘Candidatus Thiothrix sulfatifontis’ sp. nov. KT.
The complete genome of Thiothrix sp. KT was assembled from the metagenome. Results revealed a 93.93–99.72% 16S rRNA gene sequence identity to other Thiothrix species. The average nucleotide identity and digital DNA-DNA hybridization showed that the genome designated KT represents a new species within the genus Thiothrix, ‘Candidatus Thiothrix sulfatifontis’ sp. nov. KT.
In these studies, the complete genome sequence of strain CT3 was also determined. Here, because the nucleotide identity between CT3 and other Thiothrix species was below 82%, and dDDH values were less than 40%, authors concluded that the strain CT3 belonging to a novel species, Thiothrix winogradskyi sp. nov. Genome analysis showed that both strains are chemoorganoheterotrophs, chemolithotrophs (in the presence of hydrogen sulfide and thiosulfate), and chemoautotrophs, with anaerobic growth in the presence of thiosulfates.
The paper is interesting and well written, but the Chapter entitled Conclusions missing. I recommend authors to formulate and write the Conclusions of this article
Author Response
Dear Reviewer,
We have revised the article in accordance with your recommendations. All corrections made to the text are highlighted in red. Also in the responses to the corresponding corrections, the lines are indicated.
Reviewer
Comments and Suggestions for Authors
The unique and unusual habitats usually represent the main factor for the formation of new species of microorganisms. This is the case of sulfidic spring "Serovodorodny" (Tatarstan, Russia), from where were isolated and analyzed the metagenome of members of the genus Thiothrix.
The aim of this study was to determine the complete genome of Thiothrix sp. CT3 and complete sequence of MAG of Thiothrix sp. KT from bacterial fouling formed at the outflow of water from the "Serovodorodny" (Сероводородный) spring in Tatarstan, Russia (54°30’22.9"N52°09’29.4"E).
Based on pieces of information obtained from genome sequences, the authors described two new species of the genus Thiothrix, Thiothrix winogradskyi sp. nov. CT3 and ‘Candidatus Thiothrix sulfatifontis’ sp. nov. KT.
The complete genome of Thiothrix sp. KT was assembled from the metagenome. Results revealed a 93.93–99.72% 16S rRNA gene sequence identity to other Thiothrix species. The average nucleotide identity and digital DNA-DNA hybridization showed that the genome designated KT represents a new species within the genus Thiothrix, ‘Candidatus Thiothrix sulfatifontis’ sp. nov. KT.
In these studies, the complete genome sequence of strain CT3 was also determined. Here, because the nucleotide identity between CT3 and other Thiothrix species was below 82%, and dDDH values were less than 40%, authors concluded that the strain CT3 belonging to a novel species, Thiothrix winogradskyi sp. nov. Genome analysis showed that both strains are chemoorganoheterotrophs, chemolithotrophs (in the presence of hydrogen sulfide and thiosulfate), and chemoautotrophs, with anaerobic growth in the presence of thiosulfates.
The paper is interesting and well written, but the Chapter entitled Conclusions missing. I recommend authors to formulate and write the Conclusions of this article
RE: We have added a Conclusions section to the text of the article under paragraph 5 in lines 403-412.
Kind regards,
Margarita Grabovich
Reviewer 2 Report
It was very interesting because I had tried the same analysis as the authors. I thought that you had discovered a new species and its biological characteristics had been well reported.
Since the first sentences of Chapters 2.4 and 2.5 are exactly the same, I thought it would be better to omit them or change them to different expressions.
In Chapters 2.4 and 2.5, DNA was isolated from the microbial fouling of the "Serovodorodny" spring using a DNeasy PowerSoil DNA isolation kit (Qiagen, Hilden, Germany), but it seemed that the fouling of the "Serovodorodny" spring contained various impurities, and it might be difficult to isolate the DNA. If you had any pretreatment before using the DNA isolation kit, you should explain it.
In Chapters 2.3 and 2.5, the genomic DNA isolated from Thiothrix sp. CT3 and the DNA isolated from the microbial fouling of the "Serovodorodny" spring were genome sequenced using Illumina and Oxford Nanopore platforms, I thought I should explain the benefits of using the Oxford Nanopore platform (MinION instrument) for genome sequencing. It was thought that sufficient information could be obtained by using the Illumina MiSeq instrument for genome sequencing, and it had not been necessary to use the Oxford Nanopore platform (MinION instrument).
Author Response
Dear Reviewer,
We have revised the article in accordance with your recommendations. All corrections made to the text are highlighted in red. Also in the responses to the corresponding corrections, the lines are indicated.
Reviewer
Comments and Suggestions for Authors
It was very interesting because I had tried the same analysis as the authors. I thought that you had discovered a new species and its biological characteristics had been well reported.
Since the first sentences of Chapters 2.4 and 2.5 are exactly the same, I thought it would be better to omit them or change them to different expressions.
RE: We have changed the expression in paragraph 2.5. on "The total DNA was extracted from microbial fouling of the "Serovodorodny" spring (Tatarstan, Russia) as described in point 2.4." in line 148-149.
In Chapters 2.4 and 2.5, DNA was isolated from the microbial fouling of the "Serovodorodny" spring using a DNeasy PowerSoil DNA isolation kit (Qiagen, Hilden, Germany), but it seemed that the fouling of the "Serovodorodny" spring contained various impurities, and it might be difficult to isolate the DNA. If you had any pretreatment before using the DNA isolation kit, you should explain it.
RE: We did not experience any particular difficulties in isolating DNA from the microbial fouling. There may have been significant losses, but we isolated several micrograms of DNA for sequencing. No special pretreatment was used.
In Chapters 2.3 and 2.5, the genomic DNA isolated from c and the DNA isolated from the microbial fouling of the "Serovodorodny" spring were genome sequenced using Illumina and Oxford Nanopore platforms, I thought I should explain the benefits of using the Oxford Nanopore platform (MinION instrument) for genome sequencing. It was thought that sufficient information could be obtained by using the Illumina MiSeq instrument for genome sequencing, and it had not been necessary to use the Oxford Nanopore platform (MinION instrument).
RE: The use of Oxford Nanopore platform for assembly of complete genome of Thiothrix sp. CT3 was important. Assembly of only short Illunina reads only yielded more than a dozen of contigs due to the presence of repeats. In particular, the CT3 genome contains 5 copies of the 16S-23S-5S rRNA operon. The use of long Nanopore reads (with an average length of 5054 nt) allowed to “cross” these repeats and to assemble complete closed circular chromosome sequence. We added an explanation in section 2.3 - ”Since Thiothrix species usually contains multiple copies of rRNA operons, in order to obtain complete closed genome, genomic DNA was additionally sequenced on a MinION device (Oxford Nanopore Technologies, UK) using the ligation sequencing kit 1D and FLOMIN110 cells”.
Kind regards,
Margarita Grabovich